# Synucleinopathy in Amyotrophic Lateral Sclerosis: A Potential Avenue for Antisense Therapeutics?

**DOI:** 10.3390/ijms23169364

**Published:** 2022-08-19

**Authors:** Bradley Roberts, Frances Theunissen, Francis L. Mastaglia, P. Anthony Akkari, Loren L. Flynn

**Affiliations:** 1The Perron Institute for Neurological and Translational Science, Nedlands, WA 6009, Australia; 2Centre for Molecular Medicine and Innovative Therapeutics, Murdoch University, Murdoch, WA 6150, Australia; 3Centre for Neuromuscular and Neurological Disorders, University of Western Australia, Crawley, WA 6009, Australia; 4Division of Neurology, Duke University Medical Centre, Duke University, Durham, NC 27708, USA; 5Black Swan Pharmaceuticals, Wake Forrest, NC 27587, USA

**Keywords:** amyotrophic lateral sclerosis, motor neuron disease, alpha-synuclein, synucleinopathies, neurodegeneration

## Abstract

Amyotrophic lateral sclerosis (ALS) is the most common adult-onset motor neuron disease classified as both a neurodegenerative and neuromuscular disorder. With a complex aetiology and no current cure for ALS, broadening the understanding of disease pathology and therapeutic avenues is required to progress with patient care. Alpha-synuclein (αSyn) is a hallmark for disease in neurodegenerative disorders, such as Parkinson’s disease, Lewy body dementia, and multiple system atrophy. A growing body of evidence now suggests that αSyn may also play a pathological role in ALS, with αSyn-positive Lewy bodies co-aggregating alongside known ALS pathogenic proteins, such as *SOD1* and TDP-43. This review endeavours to capture the scope of literature regarding the aetiology and development of ALS and its commonalities with “synucleinopathy disorders”. We will discuss the involvement of αSyn in ALS and motor neuron disease pathology, and the current theories and strategies for therapeutics in ALS treatment, as well as those targeting αSyn for synucleinopathies, with a core focus on small molecule RNA technologies.

## 1. Amyotrophic Lateral Sclerosis

### 1.1. Epidemiology

Amyotrophic lateral sclerosis (ALS) is a fatal neurological disease characterised by the progressive degeneration of both upper and lower motor neurons [1]. Disease onset typically begins in mid- to late-adulthood, with a mean age of onset between 50 and 65 years [2], however, symptoms have been observed in patients a decade or two younger, with some patients carrying mutations that result in symptomatic onset as early as their second decade [3]. Progressive loss of motor neurons and muscle innervation results in severe muscle atrophy, weakness, and paralysis in ALS patients [4]. Further disease progression leads to the failure of the respiratory muscles, limiting patient survival to 2–5 years following symptomatic onset [5]. 

Amyotrophic lateral sclerosis is the most common subtype of adult-onset motor neuron disease with the projected disease incidence expected to increase from ~223,000 cases worldwide in 2015 to ~377,000 by 2040 [6]. In over 60 years of recorded disease history, only two drugs have received United States Food and Drug Administration (FDA) approval for the treatment of ALS, and there have been more than 70 failed drug trials in the last 20 years [7,8]. Although treatment success has thus far been limited, alternative approaches to treatment, such as the use of RNA-based therapeutics, hold great promise for the future of ALS treatment.

### 1.2. Genetics of ALS

The current classification of ALS is into two categories, inherited familial ALS (fALS) and sporadic (s)ALS, the latter developing in those patients with no known hereditary background of the disease [9]. The presentation of fALS and sALS are indistinguishable clinically, though symptoms usually present themselves earlier in familial cases [10]. Whilst ~90% of ALS cases are sporadic in nature, approximately 10% of cases are associated with the inheritance of familial gene mutations [11]. Since the discovery of the antioxidant enzyme superoxide dismutase 1 (*SOD1*) and its genetic link to ALS, there has been significant progress in understanding the genetic elements of fALS, with ~70% of the genetic mutations having been identified, linked to over 40 genes [12]. However, whilst over two-thirds of genetic mutations influencing fALS are known, the genetic contributions to sALS remain essentially undetermined, with currently only 5–10% of sALS cases being explained by mutations, leaving ~90% of patients with an unknown genetic aetiology [11,13,14]. While this may at first suggest that the genetic components of sALS play only a minor role in ALS pathology, twin studies have estimated a genetic contribution to the disease of up to 65% [15,16]. It is therefore likely that a substantial component of the heritability of sALS still remains unidentified and may lie within under-characterised regions of the human genome [17]. Variable sequences within the non-coding regions of the genome, known as structural variants (i.e., short tandem repeats), have been reported to influence the binding of regulatory elements that can alter gene expression and disease pathology [18]. Therefore, understanding the role of structural variants in the human genome will provide greater insight into how gene expression can alter disease aetiology, duration, and progression in complex diseases such as ALS [19]. One such example of a structural variant in ALS is the hexanucleotide repeat expansion in the *C9orf72* gene. The hexanucleotide motif (GGGCC) is usually repeated up to 30 times in healthy individuals but can be expanded to hundreds or thousands of segments and are a recognised cause of fALS and sALS [20]. Further understanding and characterisation of these structural variant regions will not only provide insight into mechanisms of disease modification and or disease pathogenesis, but also identify targetable areas of the genome for therapeutic intervention [12]. Current research strategies utilising antisense therapeutics to target structural variant regions, such as the *C9orf72* hexanucleotide repeat will be discussed in Section 5.2.

### 1.3. Clinical Presentation

There is no universal clinical presentation for ALS due to symptomatic progression differing greatly between patients. Typically, two-thirds of all ALS cases will have spinal-onset ALS leading to progressive weakness of the peripheral skeletal muscles, accompanied by fasciculations and muscular atrophy [21]. Patients with spinal-onset ALS initially present with symptoms such as foot drop, difficulty walking, loss of manual dexterity, and weakness when lifting the arms [22]. Symptoms may decline further in the later stages of disease progression, resulting in loss of the ability to walk and increased risk of falls, with patients becoming reliant on walking appliances and the help of caregivers [2]. A subgroup of patients with thoracic-onset ALS presents with weakness of the respiratory muscles [23]. The remaining one-third of patients will have bulbar-onset ALS, with symptomatic onset relating to the degeneration of neurons within the brainstem [24]. Common clinical presentations resulting from degeneration of bulbar neurons include dysarthria (weakness of muscles used for speech), dysphagia (difficulty swallowing), and the impaired ability to control the muscles of the throat and tongue [25]. Further degeneration of bulbar neurons may result in the inability to hold the head up (“dropped head”), whilst completely losing the ability to speak and chew [26]. Though ALS disease pathology may initially present itself in a varying manner, focal spread to the respiratory muscles and subsequent failure of the respiratory system is the most common cause of morbidity and mortality in all forms of ALS [27]. The varying progression and debilitating nature of ALS profoundly impacts not only the lives of patients, but also the lives of family members and their caregivers [28,29]. 

Originally considered a neuromuscular disorder, non-motor symptoms such as cognitive deficit and behavioural impairment were previously overlooked in people with ALS [30]. Studies in recent years have identified strong associations between disease progression and cognitive impairment in greater than 55% of ALS patients, with ~40% of patients displaying behavioural changes, specifically apathy, executive dysfunction, and disinhibition [31]. Thus, ALS research will benefit greatly from a shift in focus from a neuromuscular disorder to a classification as a spectrum disorder, with cognitive and motor symptomatic progression, similar to the symptomatic progression of other neurodegenerative disorders such as Alzheimer’s disease (AD), Parkinson’s disease (PD), and frontotemporal dementia [32,33,34]. Due to this overlap of clinical symptoms, it may be difficult to accurately diagnose people with ALS and it is not uncommon for misdiagnosis to occur in the early stages of symptomatic progression [35]. Consequently, the diagnosis of ALS typically requires evident symptomatic progression and regional spread of motor involvement over time, with the parallel presence of pathological biomarkers. Clinical diagnosis of ALS has historically used either the revised El Escorial or the Awaji criteria, however with diagnostic sensitivity lacking, new criteria were required to implement these tools into clinical practice [36]. The Gold Coast criteria, proposed in 2020, simplifies ALS diagnosis into a dichotomy, ALS or not ALS [37,38]. Through this method, the Gold Coast criteria are expected to correct sensitivity issues from past criteria, as well as facilitate early diagnosis in clinical practice [39]. Following diagnosis, the ALS Functional Rating Scale (ALSFRS) or the ALSFRS-revised (ALSFRS-R) have been the gold standard measure for monitoring the rate of change in a patient’s physical function over time [40]. However, recent clinical use of the ALSFRS questionnaires has noted an underestimation in the progressive rate of disease symptoms, and it’s not uncommon for patients to experience small reversals or a plateau in ALSFRS within a short follow-up period [41,42]. Assumed to be due to the subjective nature of the questionnaire methodologies, conclusions have highlighted the need for simultaneous measurements of more than one functional outcome as an alternative to survival-based-endpoints [43,44].

## 2. Molecular Mechanisms

The pathophysiology of ALS is multifaceted with several intraneuronal causative mechanisms suggested to play a role in disease pathogenesis, not exclusive to what is shown in Figure 1. Further understanding of the molecular mechanisms underlying motor neuron degeneration will greatly increase the probability of developing effective therapies for the treatment of ALS. These pathological mechanisms of ALS have been extensively reviewed [45] and will be discussed briefly in the present review.

### 2.1. Oxidative Stress and Mitochondrial Dysfunction

Oxidative stress is the result of an imbalance between the production and breakdown of reactive oxygen species (ROS) [46]. Superoxide radicals and hydrogen peroxide are just two of many ROS produced as metabolic by-products. As the concentration of ROS increases, it is the role of antioxidants to break down excess build-up and reduce the risk of harmful effects on vital cellular structures such as proteins, lipids, and nucleic acids [47]. Mitochondria are responsible for the production of the majority of ROS as a by-product of cellular respiration, and homeostatic dysfunction [48]. The build-up of ROS results in the progression of neurodegeneration and has been closely linked to PD and AD and is also believed to play a key role in the pathogenesis of ALS [49,50]. In *SOD1*-linked ALS, mitochondrial oxidative stress is caused by the production of mutant *SOD1* [51]. 

Mutations in the gene encoding *SOD1* account for ~2% of ALS, yet overwhelming evidence suggest that misfolded wildtype *SOD1* plays a critical role in the pathogenesis of the majority of ALS cases [52,53]. Numerous downstream effects, secondary to aggregated *SOD1*, exacerbate disease pathology, including the upregulation of ROS production and subsequent oxidative stress within the mitochondria [54,55,56]. In addition to mitochondrial oxidative stress, oxidative damage to RNA species is found in both mutant *SOD1* mouse models, and in brain and spinal cord biopsies from ALS patients [57].

### 2.2. Glutamate Excitotoxicity

Glutamate is the primary excitatory neurotransmitter essential for normal healthy brain function in mammalian nervous systems [58]. Glutamate is synthesised and stored within synaptic vesicles in the presynaptic terminal and released into the synaptic cleft where it binds to receptors on the postsynaptic terminal [59]. Within the synaptic cleft, unbound glutamate is removed by numerous cell transporter proteins, including excitatory amino acid transporters (EAATs) [60]. Motor cortex and spinal cord biopsies of both mutant *SOD1* mouse models and ALS patients have shown a reduction in EAAT homologues, leading to high levels of extracellular glutamate, overstimulation of glutamate receptors, and overall excitotoxic neuronal degeneration [2]. Furthermore, the increased concentration of extracellular glutamate causes an excessive influx of calcium ions, thought to initiate various cascades of biochemical processes known to be involved in the pathogenesis of ALS [61,62]. This pathological pathway is the target for riluzole, the first FDA-approved drug for the treatment of ALS [63].

### 2.3. Dysfunction of Axonal Transport

Axonal transport is the active process of the transference of intracellular cargo from one end of an axon to the other in order to maintain cell function and survival [64]. Neurons transfer biomaterials such as organelles bidirectionally along microtubules, from both the cell body to the synaptic junction (anterograde transport) and from the synaptic junction back to the cell body (retrograde transport). When transport is compromised, biomaterials are likely to accumulate within the neuron, causing axonal swelling, loss of cell signalling, and eventual cell death. Impairments in axonal trafficking have been closely linked with various neurodegenerative disorders, including ALS [65]. Motor neuron axons of patients with ALS have been observed to accumulate large quantities of phosphorylated neurofilament proteins and organelles, resulting in swelling along the length of the axons [66,67]. Though the exact mechanisms behind axonal transport dysregulation are unknown, research has found both anterograde and retrograde transport hindered in the presence of mutant *SOD1* protein in isogenic mouse models prior to the development of ALS-like symptoms [68,69].

### 2.4. Protein Misfolding and Aggregation

Protein misfolding is a central characteristic of most neurodegenerative disorders, yet the role of protein aggregates in ALS still remains undefined and debated [70]. Protein autophagy is an important neuroprotective pathway that is responsible for maintaining a homeostatic cellular environment [71]. Physiological events such as genetic mutations, translational errors, and cellular stress can result in protein phosphorylation and misfolding, leading to the aggregation of protein compounds within the cell cytoplasm [72]. These oligomerised and aggregated proteins are unable to be digested by proteasomal degradation and accumulate within neurons, causing an imbalance in the intraneuronal environment. With the most common genes associated with ALS pathology giving rise to misfolded protein aggregations (*SOD1*, *TARDBP*, *C9orf72*, and *FUS*), toxic protein aggregates are fundamental in the pathophysiology of ALS [73]. Questions remain surrounding the exact role of these protein aggregates in disease pathology and their toxicity in ALS; however, research continues to investigate novel protein involvements impacting ALS disease progression. Aggregations of further proteins linked to other neurodegenerative diseases, such as the PD-linked alpha-synuclein (αSyn) and the AD-linked Tau, have been identified in ALS patient motor neurons, suggesting a role for these proteins in ALS pathology. This review, however, focuses on αSyn and the current literature surrounding its involvement in ALS pathogenesis.

## 3. Alpha-Synuclein

### 3.1. Regulatory and Deregulatory Events

Alpha-synuclein is a synaptic protein found in abundance in the human brain and spinal cord, whilst also being found in the heart, liver, and other peripheral tissues of the human body [74]. Extensive research has focused on the regulatory role of αSyn on synaptic function, with knockout studies in *Drosophila* targeting the αSyn protein-encoding gene, *SNCA,* highlighting its role in vesicular trafficking, docking, and subsequent neurotransmitter release [75,76]. Although believed to maintain homeostatic function under normal expression [77], αSyn is most often referred to in association with the pathology of neurodegenerative disorders under the umbrella term “synucleinopathies”, including PD and multiple system atrophy (MSA) [78,79,80,81,82]. Furthermore, aggregations of insoluble αSyn deposits are highly associated with idiopathic PD and are unequivocally the key contributors to the formation of Lewy bodies and Lewy neurites, the hallmark inclusions in PD and in other synucleinopathies such as Lewy body dementia (LBD) [83,84]. 

Lewy bodies have long been associated with neurodegeneration and subsequent disorders and are known to have a significant pathological influence on cellular mechanisms [85,86]. Abnormal accumulation and aggregation of αSyn-dense Lewy bodies have been shown to promote the phosphorylation and aggregation of other protein compounds [87], in addition to the production of ROS [88], free radicals [89], and downstream processes of mitochondrial dysfunction [90]. Of particular interest, recent investigations have noted the colocalisation and aggregation of αSyn with the ALS hallmark protein, *SOD1* [91]. Moreover, Helferich et al., demonstrated that αSyn also accelerates *SOD1* oligomerisation independent of *SOD1* activity, suggesting common pathogenic pathways between ALS and PD neurodegeneration. In support of this theory, studies conducted in Australia have observed the accumulation of *SOD1* aggregates in degenerating regions of the post-mortem idiopathic PD brain in the absence of *SOD1* mutations, again reinforcing the overlap between ALS and PD neurodegeneration [92,93]. 

### 3.2. Genetics of Alpha-Synuclein

Mounting evidence supports the hypothesis that mutations within the *SNCA* gene facilitate protein misfolding and phosphorylation of αSyn, leading to the accumulation of toxic, insoluble Lewy body deposits, which in turn trigger a cascade of noxious events leading to neuronal degeneration (Figure 2) [94,95,96,97]. Single nucleotide polymorphisms (SNP) in the *SNCA* gene, along with overall multiplications of the gene locus and short tandem repeats, have been shown to contribute significant risk of developing Lewy body pathology, linking overexpression of *SNCA* with αSyn misfolding and pathogenic flow-on effects [98,99,100]. The first mutation identified in relation to αSyn pathology was an Ala53Thr (A53T) SNP, shown to contribute significantly to the formation of Lewy bodies and Lewy body pathology in PD [101,102]. Studies investigating the role of this SNP in relation to the development of synucleinopathy showed that transgenic mice overexpressing hSNCA^A53T^ display substantial amounts of aggregated αSyn staining in spinal motor neurons and is associated with limb paralysis and death, suggesting a pathological role for αSyn in motor neuron vulnerability [103,104].

One of the best-studied variations in the *SNCA* gene is the microsatellite, Rep1, located 10 kb upstream of the start codon in the promoter region of the gene, first described by Xia et al. [105]. A number of association studies have investigated the link between the Rep1 dinucleotide repeat region and PD risk, exploring the allelic contribution to overexpression of *SNCA,* αSyn and Lewy body pathology, though opinions on this pathological relationship remain disputed [106,107,108]. While several studies have concluded that the length of the Rep1 variant is linked to PD risk and cognitive decline [106,109], Soldner et al., found no effect of the Rep1 variant on *SNCA* expression [110]. However, a recent review noted that this study had difficulty quantifying specific *SNCA* transcript expression, casting doubt on the significance of the findings [111]. 

### 3.3. Environmental Effectors of Synucleinopathy Pathology

In the vast majority of idiopathic PD cases, environmental impacts are thought to play a critical role in the initiation and development of the disease, however, the excitement surrounding the discovery of genetic contributors in familial and sporadic forms of the disease may have diverted attention away from the environmental causes of PD and other synucleinopathies [112]. Still, epidemiological studies investigating factors affecting PD risk have shown that exposure to certain toxins and specific living conditions can increase the likelihood of disease occurrence and risk [113,114]. Chemical substances that have long been used in agricultural farming, including rotenone and 1-methyl-4-phenyl-1, 2, 3, 6-tetrahydropyridine (MPTP), have been closely observed in both in vivo and in vitro studies to significantly increase the risk of PD symptomatic onset and exacerbate disease through inducing the aggregation of αSyn protein [115,116,117]. Furthermore, MPTP and rotenone have been used in rodent models to determine neuronal pathways involved in αSyn propagation from the olfactory bulb into the midbrain following the inhalation of noxious chemicals [118]. These laboratory findings have been crucial in understanding how environmental effectors can induce a PD phenotype and have aided in furthering clinical understanding of the heightened PD occurrence in agricultural farming communities [119,120]. 

A rapidly growing body of research investigating the role of the gut microbiome and dysbiosis on synucleinopathy development has transpired over recent years; in particular the role of the gut-brain axis and the propagation of αSyn to the brain via vagal afferents [121]. Although this field is vast, one area of interest involves the naturally occurring bacterial toxin ß-N-methylamino-L-alanine (BMAA). Occasionally found in dangerously toxic concentrations within certain seafood, BMAA consumption has been shown to cause a significant shift in the microbial environment of the gut [122,123], demonstrating to have profound effects on the propagation of αSyn from the gut to the brain and the subsequent development of PD [122,124]. Interestingly, BMAA has causal links to both ALS and PD, being the key contributor to the rare incidence of the ALS-PD complex in the western Pacific islands of Guam and Japan [125,126]. The ALS-PD complex was the result of a multigenerational occurrence of neurological conditions with clinical features manifesting as ALS followed by PD, or PD followed by ALS, all intertwined with a dementia phenotype. In linking these neurodegenerative diseases, the Guam incidence has been integral in building an understanding of the mechanisms behind neurodegeneration and the role of environmental factors. Although the ALS-PD complex/BMAA hypothesis was abandoned multiple times due to studies finding minute amounts of free BMAA in the Pacific environment [127,128], the hypothesis has had a resurgence in the last decade as a result of numerous clinical, epidemiological and experimental research studies reviewed by Bradley and Mash [129]. Furthermore, an in vitro study modelling BMAA consumption has demonstrated the structural destabilisation of *SOD1* proteins following the substitution of any serine with BMAA, resulting in promotion of neurotoxic *SOD1* aggregation [130]. Similarly, BMAA misincorporation in αSyn is thought to promote protein aggregation via the destabilisation of the extension in Serine 129 phosphorylation [123], the major posttranslational modification of αSyn aggregation [131].

These important connections between the ALS/PD complex and the colocalisation and interplay between key disease proteins and genetic elements are crucial when questioning a broader link between neurodegenerative diseases and whether αSyn may play a leading role in bridging this gap.

## 4. Is Amyotrophic Lateral Sclerosis a Synucleinopathy?

Although historically considered a neuromuscular disorder, recent clinical research has investigated the non-motor symptoms of ALS, revealing that ~30% of ALS patients display Parkinsonian-like traits similar to those seen in PD and other synucleinopathies [132]. To further explore the overlap between these diseases, a handful of studies have investigated the pathological role of αSyn and Lewy bodies in ALS. 

The post-mortem observation of αSyn-positive Lewy bodies in patient neuronal cells typically correlates with a clinical diagnosis of PD or LBD and has not previously been considered a characteristic or hallmark of ALS [102]. Nevertheless, studies have shown Lewy body-like inclusions in the anterior horn of the spinal cord and bulbar regions of fALS patients, in the absence of inclusions within the midbrain areas typically affected in PD [133]. Similarly, significant numbers of immunoreactive αSyn protein inclusions have been noted in the spinal cords of ALS patients, with the characteristic morphology of the αSyn-dense Lewy bodies found in PD patients [134]. These post-mortem studies ruled out PD as an incidental secondary diagnosis in these cases, leading to speculation that αSyn is involved in ALS pathology. Furthermore, observations reported in a separate study concluded that αSyn protein aggregates in ALS patient glial cells may have a significant role in neuronal cell death and symptomatic onset similar to the mechanisms observed in MSA [135].

In vivo studies have found evidence of increased αSyn immunoreactivity in the central nervous system of hSOD1^G93A^ transgenic mice, supporting the hypothesis that the effects of αSyn in ALS have been underestimated and that αSyn protein aggregates may play a significant role in the pathogenesis of ALS [136]. In support of this theory, and as previously mentioned, multiple studies have observed a strong interplay between the αSyn and *SOD1* proteins, highlighting their influences upon each other to induce and enhance aggregational properties [91,137]. In these studies, the presence of cytoplasmic immunoreactive inclusions containing both *SOD1* and phosphorylated-αSyn were noted in both hSOD1^G93A^ mice and patient brain tissue. Moreover, a reduction in polymerisation and aggregation of αSyn was demonstrated in hSOD1^G93A^ mice treated with bee venom, a traditional medicine for inflammatory diseases used to rescue the impairment of the ubiquitin-proteasome system in ALS, further strengthening the hypothesis surrounding the relationship between αSyn and *SOD1* [138,139]. Interestingly, one further study noted a significant increase in *SOD1* oligomerisation following the introduction of misfolded αSyn preformed fibrils both in vitro and in vivo in hSOD1^G93A^ mice [140]. This study provides evidence that αSyn can induce or exacerbate aggregation of known ALS proteins, leading these authors to suggest that suppressing αSyn-mediated protein interaction may have therapeutic potential for ALS.

Recent genetic studies have taken the considerable overlap of clinical and pathological features of PD, ALS, and MSA into account, and have highlighted the importance of testing known pathological genetic mutations across these neurodegenerative diseases [141,142,143]. Collectively, these studies found no association between PD-related *SNCA* variants and ALS, however, their analyses focused on SNP-based risk association, and it has been previously shown that only ~8% of heritability in ALS is associated with genome-wide SNPs [144]. Thus, these papers do not rule out the possibility for ALS risk to be associated with those more complex regions of the *SNCA* gene linked with PD pathology.

Clinical documentation of patients in post-mortem research has also revealed the coexistence of PD and ALS symptoms in patients who exhibited αSyn Lewy body pathology in the brainstem in conjunction with TAR DNA-binding protein 43 (TDP-43) immunoreactive deposits in the motor nuclei [145,146]. These studies suggest that TDP-43, a pathological hallmark in ALS, and αSyn may also play a synergistic role in neurodegenerative disease pathogenesis [147,148]. A more recent study investigating the synergistic role of TDP-43 and αSyn found that ~15% of all TDP-43 positive ALS cases also displayed significant Lewy body pathology [149]. Similarly, co-existing TDP-43 pathology was noted in 31% of cases of LBD with AD, 7.2% of PD cases, and 19% of PD cases with dementia [150], further highlighting the overlap between ALS pathology and synucleinopathies.

### Alpha-Synuclein in the Motor Neuron

As previously stated, αSyn is a synaptic protein involved in membrane vesicle trafficking and cytoskeletal dynamics in neuronal cells [77]. Due to the likely role of αSyn in dopamine regulation [151], dopaminergic neurons are the most vulnerable in PD and thus are the most well-studied neuronal subtype in association with neural pathology of PD [152]. However, with an undisputed regulatory role in other neuronal subtypes, the accumulation of toxic αSyn bodies can lead to the degeneration of numerous classes of neurons [153]. With the literature of the last two decades dedicated to understanding the propagation of αSyn to the brain, interest has also been drawn to the effects of αSyn on different neuronal subtypes in the spinal cord, with studies observing Lewy body-like inclusions in spinal motor neurons of hSNCA^A53T^ transgenic mice [104,154]. These were not, however, the first observations of αSyn in motor neuron pathology. Studies in the early 2000s noted that although αSyn and ubiquitin-positive inclusions were observed in structures classically affected in synucleinopathies, most of the pathological changes observed in the spinal cord were within the spinal motor neurons, irrespective of any modifications in the nigrostriatal system [155,156].

More recently, immunohistology research in spinal cord injury noted that changes in *SNCA* regulation could alter the differential survival of spinal cord neurons, with up-regulation of *SNCA* resulting in neuronal apoptosis [157]. Similarly, αSyn has been shown to accumulate in various regions of the human spinal cord tissue, with Lewy bodies and Lewy neurites observed in large motor neurons of the lower thoracic and lumbar spinal cord, correlating with clinical features of lower limb denervation in idiopathic PD [158,159,160]. Interestingly, MPTP has not only been demonstrated to be toxic to dopaminergic neurons with detrimental effects in PD pathogenesis, but the neurotoxin also induces the death of motor neurons in vivo, thus bridging PD and ALS pathology [161].

When comparing the role of αSyn in neuronal cell pathology in the brain and spinal cord, one study carried out in hSNCA^A53T^ transgenic mice observed a deleterious influence of αSyn in the spinal cord, with overexpression of αSyn being associated with severe pathological impairments in glial cells of the spinal cord, with no loss of neurons in the brain [162]. Glial cells have been shown to play a key role in neurodegenerative diseases, with astrocytes having an essential function in the degeneration of spinal motor neurons in ALS [163], and oligodendrocytes actively contributing to neuronal death in MSA [164]. However, in the infant-onset ALS-like disease, spinal muscular atrophy overexpression of αSyn was shown to be neuroprotective against neuron loss, thus implying that a controlled regulation of αSyn expression is vital to the overall health of neuronal subtypes, and in particular, to motor neurons [165,166].

The studies discussed above confirm the presence of αSyn and Lewy body-like deposits in individual cases of ALS and suggest a pathological role for αSyn expression in motor neuron degeneration. However, whether αSyn contributes to ALS pathology by exacerbating disease mechanisms, including mitochondrial dysfunction and oxidative stress, or whether αSyn and Lewy body pathology are a result of cellular processes associated with ALS disease is yet to be determined.

## 5. Current Treatment

### 5.1. Current Therapeutic Options for ALS

With no current cure for the disease, ALS therapeutics have historically centred around attempting to slow the progression of the disease and providing symptomatic treatments to maintain and improve patient quality of life [167]. Pain management is integral to improving the quality of life, with a recent meta-analysis estimating a pooled prevalence of 60% of patients suffering from severe pain symptoms [168]. Therapeutic management of pain includes the use of non-steroidal anti-inflammatory drugs and opioids, with alternative therapies, such as medicinal cannabis, occasionally used to treat severe pain symptoms [169]. Therapeutic exercise and physiotherapy are also recommended for patients to slow symptomatic progression and alleviate pain related to progressive muscle weakness and atrophy [170].

In the later stages of disease progression when the bulbar muscles deteriorate, interventions to manage weight loss and malnutrition are fundamental in prolonging patient survival [171]. Multidisciplinary teams involving dieticians, occupational therapists, speech pathologists, and physical aids to assist in respiratory therapy, are widely recognised to improve patient quality of life and when combined with medicinal therapeutics, have the potential to prolong patient survival [172,173].

In the 60 years of ALS disease history, there have only been two drugs approved by the FDA, with more than 70 randomised drug trials in ALS patients showing a little beneficial effect on patient survival [174]. Riluzole, the most widely used and only treatment available in Australia, was approved in 1995 and initially hypothesised to act as a glutamate antagonist, inhibiting the accumulation of glutamate at the synaptic junction and thereby reducing glutamate excitotoxicity [45]. However, subsequent research has shown that it has only limited effects on glutamate receptors and has suggested a more complex mechanism of action [175]. Original reports of randomised clinical trials showed an estimated increase in patient survival of up to 2–3 months [176], though some more recent studies have suggested that riluzole may increase survival by up to 6–19 months [63]. Although its efficacy remains controversial, riluzole has proven to be effective when used in combination drug therapies [177].

The second drug, edaravone, was approved in the US and Japan in 2017 and is an antioxidant compound, proposed to reduce oxidative stress and eliminate lipid peroxidation, although the exact mechanism of action remains unknown [178]. Initially approved for the treatment of cerebral embolism and stroke, edaravone was observed to have a therapeutic effect in ALS patients, with treated patients exhibiting reduced functional loss compared to patients who received a placebo [179,180]. However, recent studies on the efficacy and safety of edaravone in an ALS subpopulation reported little to no significant beneficial effects, with the study failing to demonstrate the efficacy of edaravone to slow the progression of the disease [181,182]. 

The poor efficacy and inconsistency in the mechanisms and therapeutic potential of the current FDA-approved treatments for ALS indicate that new strategies are urgently needed to formulate therapeutic development for patient treatment. To date, most postulated treatments for ALS have investigated small molecules that are theorised to limit the progression of ALS pathological pathways, however, there is hope that more personalised experimental treatments may prove more successful in prolonging patient survival. Advancements in RNA technologies, such as antisense oligonucleotide (AO) therapies, offer encouraging methods for the development of personalised medicine, directly targeting and modulating genes known to be involved in an individual’s disease pathology [183].

### 5.2. Pipeline Development for Antisense Therapies in ALS

Genetic therapies have begun to demonstrate therapeutic potential in not only monogenic neurological disorders but also in complex neurological disorders involving numerous defective genes [184]. Antisense oligonucleotides are one method used to alter the expression of target proteins via the modulation of RNA processing and translation [185]. Whilst there are currently no AO therapies approved for use in ALS treatment, three promising AO drug strategies have entered clinical development to directly target *SOD1*, *C9orf72*, and *FUsed in Sarcoma (FUS)* expression in patients carrying mutations in these genes, and one antisense drug targeting *ATXN2* is currently in clinical evaluation for those with sporadic disease [186]. This section will discuss the novel targets currently under investigation for potential AO development, and the available clinical data of molecules targeting both *SOD1* and *C9orf72* in ALS clinical trials.

A recurrent hallmark of ALS pathology is the presence of cytoplasmic inclusions containing TDP-43 observed in histological sections of both the cerebral cortex and spinal cord [187,188]. With over 90% of patients displaying aggregations of TDP-43 in motor neurons, there is substantial biological evidence to support the therapeutic potential for AOs targeting the knockdown of TDP-43 [189]. TDP-43 has also been observed to play a critical role in transcription, nuclear-cytoplasmic transport, and RNA processing, with changes in RNA metabolism resulting from the loss of nuclear TDP-43 [190,191]. Due to the fluctuant nuclear and cytoplasmic expression of TDP-43 inferring both a gain of function (GOF) and a loss function (LOF) mechanism in ALS, simply reducing TDP-43 expression may not be sufficient to alleviate ALS pathology [192]. Thus to date, transcripts disrupted by the loss of nuclear TDP-43 are currently under investigation as emerging AO targets for the treatment of ALS [193]. Another mechanism presenting as an emerging strategy for TDP-43-antisense therapy is the use of AOs to restore autoregulatory control of TDP-43. TDP-43 autoregulation is the mechanism by which TDP-43 regulates its own expression in a negative feedback loop by binding and modulating the pre-mRNA splicing within the 3′ UTR of the *TARDBP* transcript [194]. A study crossing both TDP-43 GOF and LOF mouse models produced a compound heterozygote model that was shown to restore TDP-43 auto-regulation and subsequently partially rescues the phenotype, demonstrating that the LOF and GOF may compensate for each other in vivo [195]. Since both TDP-43 GOF and LOF mouse models result in the upregulation of *TARDBP* [196], this evidence suggests that leveraging this natural mechanism may provide a therapeutic avenue whereby the modulation of *TARDBP* can restore auto-regulation and break the negative feedback cycle of TDP-43 mislocalisation and aggregation.

One transcript directly influenced by the availability of TDP-43 is *STMN2*, encoding the stathmin 2 protein. Stathmin 2 is a tubulin-binding protein with a critical role in microtubule dynamics and neurite outgrowth [197]. Aggregation of TDP-43 and subsequent LOF facilitates the inclusion of a cryptic exon in the *STMN2* transcript resulting in an early stop codon and polyadenylation, reducing the total expression of stathmin 2 protein [198]. Post-mortem analysis of ALS patient spinal cord tissue showed an overall reduction in functional *STMN2* transcript expression when compared to controls, with higher expression of the truncated *STMN2*-mRNA transcripts [198,199]. In a recent in vivo study, *STMN2*-knockout mice were shown to recapitulate aspects of ALS neuromuscular pathology, providing further support for a critical role for *STMN2* expression in ALS pathology [200]. *STMN2*-knockout mice demonstrated significant distal muscular weakness and loss of neuromuscular junctions in the periphery, as well as demonstrating sensory deficits concurrent with the ALS phenotype. As such, AO strategies to restore full-length mature *STMN2* mRNA and increase overall *STMN2* expression are currently underway as a potential therapeutic for the treatment of ALS [201].

An alternative approach to overcoming TDP-43 pathology is to target the RNA-binding protein ataxin 2 (*ATXN2*), known to influence TDP-43mislocalisation and propensity to aggregate within the cytoplasm [193]. The *ATXN2* gene contains an expanded trinucleotide CAG repeat, with longer repeat lengths found to be a cause of spinocerebellar ataxia type 2 and intermediate repeat lengths found in patients with ALS [202,203]. An antisense therapeutic designed to decrease *ATXN2* expression (BIIB105) has demonstrated increased survival in a mouse model of TDP-43 ALS and is currently undergoing phase I clinical evaluation (NCT04494256) [193]. Although in the early stages, targeting the *ATXN2* transcript may therefore provide another potential drug target to modify disease progression in ALS.

With recent evidence conferring a toxic GOF mechanism in ALS caused by *FUS* aggregation, overexpression, or cytoplasmic mislocalisation, a subgroup of ALS patients with *FUS*-related pathology may be amenable to an AO designed to down-regulate *FUS* as a therapeutic approach [204]. One patient has already received an investigational *FUS*-targeted AO (ION363) and a phase I-III clinical trial for this drug has recently been announced (NCT04768972) [205]. 

The *C9orf72* targeting AO BIIB078 was recently evaluated in phase I clinical trial and designed to selectively target *C9orf72* transcripts with repeat expansions, whilst preserving the expression of *C9orf72* transcripts from the unaffected allele (NCT03626012). This approach aims to reduce the acquired toxicity from the expression of repeat-associated RNA foci, dipeptide proteins, and reduced *C9orf72* expression seen in clinical samples [206,207]. Following success in preclinical models, the BIIB078 AO entered phase I trials in September 2018 to assess drug safety and tolerability. BIIB078 was well tolerated across the treatment groups with no differences in the frequency and severity of adverse effects reported between the treatment and placebo groups. However, in March 2022, the trial was terminated due to BIIB078 not meeting primary or secondary endpoints, displaying no consistent differences in clinical benefits, including ALSFRS-R score, between the low dose treatment and placebo groups [208]. Interestingly, according to the Biogen announcement, patients receiving the higher dose of BIIB078 trended towards a greater decline when compared to placebo, and based on these data, the study was discontinued [208]. Alternative AO strategies and sequences targeting *C9orf72* are currently undergoing clinical evaluation, including WVE-004 (NCT04931862) and afinersen, which showed promise at reducing *C9orf72* in a single patient trial [209].

Following the discovery that *SOD1*-knockout mice fail to develop motor neuron loss [210], it was hypothesised that reducing the expression of *SOD1* may prove feasible as an ALS therapeutic strategy [184]. Tofersen (BIIB067) is an antisense therapy designed to target and reduce *SOD1* mRNA, currently under evaluation in phase III studies for patients carrying pathogenic *SOD1* mutations. A phase I/IIa study of tofersen was completed in 2020, demonstrating up to 36% reduction in *SOD1* levels within the cerebrospinal fluid (CSF) and showed promise at slowing disease progression when compared to the placebo group, with patients receiving tofersen experiencing an improvement in ALSFRS-R measures [211]. However, primary safety concerns were reported in those receiving the drug, including CSF pleocytosis and increased CSF protein levels, yet these observations did not meet the definition of adverse events. Based on these encouraging results, tofersen has progressed into phase III clinical studies (NCT02623699). In this subsequent phase III study, adverse effects following the administration of tofersen were again reported, including myelitis with sensory and motor deficits [212]. In October 2021, Biogen announced that tofersen failed to meet the primary endpoint of the phase III VALOR trial, with no significant change in ALSFRS-R between treatment and placebo groups [213]. Although the primary efficacy endpoints of this study were not reached, some positive results were observed in key secondary endpoints; administration of tofersen induced sustained reduction in CSF *SOD1* levels, demonstrating positive target engagement, as well as the potential to reduce neuronal damage, supported by reduced plasma CSF levels [213]. Long term clinical efficacy of tofersen is currently being investigated in an open-label extension study (NCT03070119).

Although in various stages of development, the results from the above studies provide promise that more effective disease-modifying therapies for ALS patients, such as AOs, will soon become available. While AO therapies have shown effective target modification and an improvement in clinical biomarkers, failed primary outcomes suggest a need for improved clinical trial design, highlighting the challenges of readouts such as the ALSFRS-R rating, and the opportunity for using dynamic clinical biomarkers, such as neurofilament light, as alternative primary endpoints [42]. Further, modifications to improve the safety profile of existing AO chemistries, as well as the development of novel AO chemistries, will likely overcome drug-related adverse events and improve clinical trial outcomes [214]. In the meantime, perhaps broadening the scope to more novel targets may prove fruitful when designing innovative therapeutic techniques for neurodegenerative diseases.

### 5.3. Current Development and the Future of αSyn-Targeted Therapies

Alpha-synuclein has been the major therapeutic target for PD and other related synucleinopathies for the past three decades, but no cure is yet available for disorders involving toxic αSyn. With extensive research describing the association between increased levels of αSyn and neurodegenerative diseases, a number of hypotheses proposing methods for reducing cellular levels of αSyn as a possible approach to therapy have arisen [215].

Below is a brief summary of approaches for reducing αSyn levels and subsequent toxicity to alleviate neuropathology. 

#### 5.3.1. Reducing Uptake of Extracellular Alpha-Synuclein

Strategies to inhibit the propagation of toxic αSyn are still in the early theory and developmental stages. Through a recent comprehensive series of experiments, αSyn preformed fibrils were observed to bind to lymphocyte-activation-gene-3 (LAG3) proteins on the post-synaptic cell membrane with a high affinity [216]. Mao et al., describe that the binding of αSyn to LAG3 receptors initiates the uptake of pathological αSyn into neurons, transporting the toxic protein from cell to cell. Antibodies specifically designed to bind to LAG3 showed a decrease in αSyn uptake and propagation, demonstrating that blocking or eliminating LAG3 functionality may be a therapeutic target in regulating the propagation of pathological αSyn [216].

Though naïve in its development, and whilst there may be other proteins involved in the propagation of toxic αSyn [217], the Mao et al., study demonstrated that subsequent to the development of Lewy body pathology, intercellular αSyn propagation may be inhibited to alleviate the pathological progression of synucleinopathy disorders. 

#### 5.3.2. Increasing Extracellular Alpha-Synuclein Degradation

Active and passive immunotherapies targeting extracellular αSyn are some of the technologies at the forefront of αSyn targeted therapies for synucleinopathies. Active immunisation involves the stimulation of the immune system to produce antibodies against target proteins, whilst passive immunisation involves the direct administration of targeted antibodies designed to protect against specific disease pathology. Currently, in phase II clinical trials, Biogen’s passive immunotherapy using the αSyn-targeted antibody BIIB054 was proven to be well tolerated in healthy participants, with detectable levels of the antibody in patient CSF (NCT03318523) [218]. Although the data supported the continued development of BIIB054, the project was discontinued in February 2022 due to the failure of the study to meet both the primary and secondary endpoints.

#### 5.3.3. Promoting Degradation of Intracellular Alpha-Synuclein Aggregates

Autophagy is believed to play a major role in the degradation and clearing of pathological αSyn aggregates [219], and thus, theoretically enhancing αSyn autophagic processes will enhance the clearing of pathological αSyn. A class of drugs related to the autophagy inducer rapamycin and rapamycin analogues has been shown to increase macroautophagic function, resulting in the clearance and reduction in αSyn aggregation and subsequent cellular toxicity [220]. However, a lack of specificity and the occurrence of adverse events have limited the potential use for such drugs where long-term administration would be required to target synucleinopathy pathology. 

Many alternative strategies are in development for promoting αSyn degradation, including the exciting approach to increase the expression, stability, or delivery of the lysosomal enzyme ß-gluccocerebrosidase (GCase). Mutations in the *GBA1* gene, encoding GCase, have been shown to confer a 20-30-fold increase in the risk of developing PD [221]. These mutations have been shown to reduce the activity of GCase in both PD brain tissue and CSF with a reported correlation between the loss of GCase function and accumulation of aggregated αSyn [222,223,224]. These strong genetic links have made GCase a prominent target for drug development in treating synucleinopathies, with two drugs moved into phase II clinical trials, ambroxol (NCT02941822; NCT02914366) and GZ/SAR402671 (MOVES-PD). Though the MOVES-PD study was terminated after not reaching primary endpoints, ambroxol therapy was safe and well tolerated and further placebo-controlled clinical trials are needed to study the effects of ambroxol on the progression of PD [225].

#### 5.3.4. Inhibiting Alpha-Synuclein Aggregation

If αSyn aggregation is a key contributor to neurodegeneration in synucleinopathies, eliminating the potentiation for αSyn to aggregate may prove therapeutic in alleviating αSyn toxicity. Two programs are currently in clinical testing phases and aim to do exactly this. First, a small molecule therapy designed in partnership with Neuropore and UCB Pharma, NPT200-11. NPT200-11 has been shown to prevent toxic misfolding and aggregation of αSyn, improving behavioural, neuropathological, and biochemical outcomes in transgenic mice overexpressing αSyn [226]. Second, NPT088 is a fusion protein designed to selectively target multiple misfolded proteins simultaneously. Pro-clara Bioscience has reported NPT088′s strong affinity for αSyn, whilst also reporting positive effects on tyrosine hydroxylase levels in an αSyn-overexpressing transgenic mouse model, an enzyme upstream in the production of dopamine [227].

#### 5.3.5. Reducing Alpha-Synuclein Production

If, as hypothesised, synucleinopathies are the result of a toxic GOF with αSyn accumulation and aggregation, inhibition of αSyn production through gene and protein knockdown has therapeutic potential for many patient populations [228,229]. While this review focuses on the use of antisense therapeutics, the repurposing of small molecule drugs to regulate *SNCA* gene transcription has also been extensively reviewed elsewhere [230]. 

To date, few studies have investigated the role of modulating αSyn through antisense therapeutics. One study noted a reduction in αSyn synthesis and the prevention of toxic accumulation of αSyn protein aggregates following the administration of an αSyn-targeting AO in mice with adeno-associated virus-mediated hSNCA overexpression [231]. More recently, a PD rodent model treated with an αSyn-targeted AO demonstrated that AO-mediated suppression of *SNCA* not only prevented further αSyn aggregation but also cleared the established αSyn pathology in a dose-dependent manner [232]. Cole et al., concluded that a reduction in *SNCA* transcript by just 50% proved beneficial in downregulating αSyn protein expression [232]. Similarly, mice with induced Lewy body pathology via injection of αSyn pre-formed fibrils and treated with an αSyn-targeted AO, showed a reduction in total αSyn and phosphorylated-αSyn protein levels by >50% after a single dose [233]. Although both studies report a successful reduction in αSyn pathology, Boutros et al., also reported deleterious off-target effects, including reduced food intake, weight loss, and impaired motor abilities in αSyn-AO treated mice, independent of the effects of lowering αSyn [233]. Ionis Pharmaceuticals are currently recruiting for a phase I safety and tolerability study of the same αSyn-AO to be evaluated in MSA patients (NCT04165486).

The above strategies to reduce overall αSyn protein production have the potential to alleviate synucleinopathy disease pathology and resultant symptoms. Furthermore, given the overlap in the causative mechanisms of ALS, PD, and other synucleinopathies, these strategies may also be implemented into therapeutic practice to alleviate the disease progression of ALS. The current literature presented in this review demonstrates that αSyn has the potential to aggregate in ALS motor neurons, however, how broadly this is observed and the overall role of αSyn in ALS pathology is yet to be determined. Further studies of ALS patient cohorts investigating αSyn expression and Lewy body pathology will help to identify the targetable population and scope of patients amenable to an αSyn knockdown therapy. If patients can be screened for synucleinopathy positive mutations, the administration of AOs during the pre-symptomatic phase, therefore, has the potential to prevent the development or progression of disease pathology, prior to the irreversible loss of neurons [234]. Therapeutics utilising methods of providing an upstream effect such as those following AO administration may be the most effective in preventing or reducing synucleinopathy pathology. Though further research is required to determine the mechanistic role of αSyn in ALS pathology, perhaps an αSyn-AO in combination therapy with an AO designed to target a known ALS hallmark will be beneficial in alleviating disease. Given the presence of *SOD1* and TDP-43 pathology in PD and of αSyn and Tau pathology in ALS, therapies targeting each of these proteins may be utilised across the broader range of neurodegenerative disorders.

## 6. Concluding Remarks

With the global prevalence of ALS expected to rise by 29% by 2040, new therapeutic strategies for the ALS community are urgently required to combat an exponentially rising disease epidemiology [6]. Currently, only symptomatic treatments are available and disease-modifying therapies have been elusive with over 70 randomised trials of putative new therapeutic agents over the past two decades, having failed to demonstrate clinical efficacy [7]. Personalised medicine approaches using RNA technologies and AO developments have revealed promising results for ALS gene-targeting therapeutics, however additional therapeutic targets and novel approaches need to be identified to expand the potential therapeutic avenues for people who have ALS or are at risk of developing the disease. 

The current application of AOs to target αSyn in rodent PD models has shown positive results in knocking down gene and protein expression with subsequent favourable behavioural, biochemical, and neuropathological outcomes. The same AO therapeutic is soon to be evaluated in the clinical for MSA patients, with the results eagerly awaited. Thus, it is not too hard to imagine that antisense therapies targeting the αSyn encoding gene, *SNCA*, could be effective and alleviate the progression of all diseases under the synucleinopathy umbrella. With a recent increase in evidence supporting the involvement of αSyn pathology in ALS disease pathogenesis and motor neuron pathology, such αSyn-targeted therapies may prove successful for those subsets of ALS patients expressing markedly higher levels of αSyn protein. Screening tools for genetic markers linked to disease, such as those structural variants associated with ALS, PD, and synucleinopathy described earlier, will be essential for identifying future clinical trial cohorts of patients likely to benefit from αSyn-targeting therapies. Thus, further research is required to provide a clearer understanding of synucleinopathy development in various ALS patient groups, at different stages of the disease process, as well as further investigation of the interactions between αSyn and other ALS-associated proteins across the spectrum of neurodegenerative diseases.

## Figures and Tables

**Figure 1 ijms-23-09364-f001:**
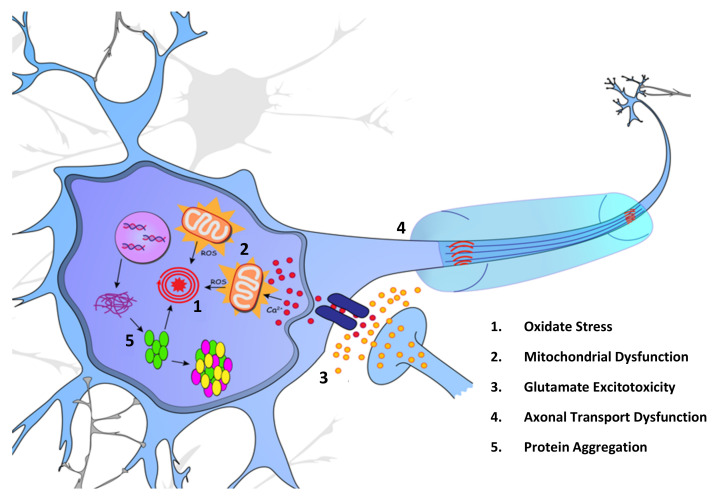
Proposed intracellular pathogenic mechanisms of ALS. (1) Oxidative stress caused by a multitude of factors, is believed to progress disease pathology in ALS patients due to the harmful side-effects of ROS build-up. (2) Build-up of reactive oxygen species (ROS) from the dysregulation of mitochondrial function has been proposed as an initiating factor for ALS. Multiple ALS-linked proteins interact with cellular mitochondria, including *SOD1*, TDP-43, and *FUS*. (3) Overstimulation of glutamate receptors has been proposed as a mechanism for ALS disease pathogenesis and is the disease pathway targeted by the first approved ALS therapy, riluzole. (4) Impaired axonal transport has been implicated in ALS disease pathology, with the presence of protein aggregations inhibiting functional transport along microtubules. (5) With the impairment of proteostasis, proteins such as *SOD1* and TDP-43 accumulate within the cytoplasm, further impeding cell health.

**Figure 2 ijms-23-09364-f002:**
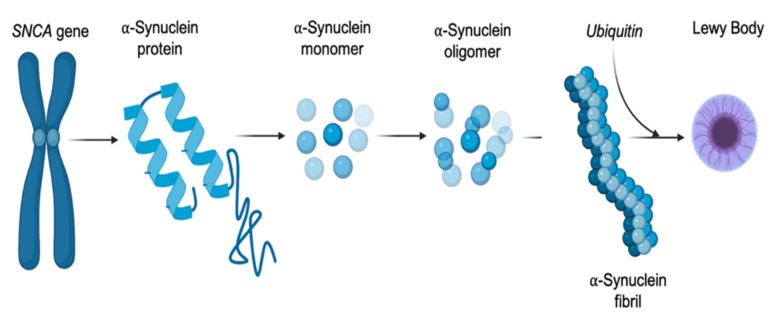
The pathway of alpha-synuclein (αSyn) dysregulation. Modification and mutations within the *SNCA* gene can alter the expression of the encoded αSyn protein, facilitating the protein down a fibrillar pathway, leading to misfolding and phosphorylation.

## Data Availability

Not applicable.

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
