# Peer review of "Synucleinopathy in Amyotrophic Lateral Sclerosis: A Potential Avenue for Antisense Therapeutics?"

_ijms, 2022, doi:10.3390/ijms23169364_

Round 1
Reviewer 1 Report
This review entitled “Synucleinopathy in amyotrophic lateral sclerosis: a potential avenue for antisense therapeutics?” by Bradley Roberts et al., provides comprehensive coverage of the molecular mechanisms and development of ALS and its commonalities with “synucleinopathy disorders”. This manuscript also discussed the involvement of alpha-synuclein in ALS and motor neuron disease pathology, and the current theories and strategies for therapeutics in ALS treatment. Especially, they put forward that AOs therapies targeting the alpha-synuclein could be effective and mitigate the progression of neurodegenerative disorders with synucleinopathy. This is considered a very novel perspective. This is a generally well-written review, the structure is good and the figure is clear. I have only minor suggestions.
1. Suggest that add some evidence about reducing É‘-Syn which can specifically alleviate cellular defects of ALS, such as oxidate stress, mitochondrial dysfunction, glutamate excitotoxicity, and axonal transport dysfunction which are mentioned in the “Molecular Mechanisms” section. Here should highlight the advantages of AOs therapies targeting the alpha-synuclein to treat ALS compared with the AOs Therapies in ALS targeting SOD1, C9orf72, and FUS.
“ ~30% of ALS patients display Parkinsonian-like traits similar to those seen in PD and other synucleinopathies”. It indicated that AOs therapies targeting the alpha-synuclein may not be applied to all ALS treatments. So this review should clarify the scope of AOs therapies targeting the alpha-synuclein for ALS treatment.
Author Response
Reviewer One
Comments and Suggestions for Authors
This review entitled “Synucleinopathy in amyotrophic lateral sclerosis: a potential avenue for antisense therapeutics?” by Bradley Roberts et al., provides comprehensive coverage of the molecular mechanisms and development of ALS and its commonalities with “synucleinopathy disorders”. This manuscript also discussed the involvement of alpha-synuclein in ALS and motor neuron disease pathology, and the current theories and strategies for therapeutics in ALS treatment. Especially, they put forward that AOs therapies targeting the alpha-synuclein could be effective and mitigate the progression of neurodegenerative disorders with synucleinopathy. This is considered a very novel perspective. This is a generally well-written review, the structure is good and the figure is clear. I have only minor suggestions.
- Suggest that add some evidence about reducing É‘-Syn which can specifically alleviate cellular defects of ALS, such as oxidate stress, mitochondrial dysfunction, glutamate excitotoxicity, and axonal transport dysfunction which are mentioned in the “Molecular Mechanisms” section. Here should highlight the advantages of AOs therapies targeting the alpha-synuclein to treat ALS compared with the AOs Therapies in ALS targeting SOD1, C9orf72, and FUS.
We thank the reviewer for this comment, and we have addressed this in the review. Further research is needed in this area in order to draw definitive conclusions as to how alpha-synuclein targeted AOs can help alleviate ALS disease pathology. Comparison with AOs targeting known ALS hallmarks is difficult and perhaps a combination therapy will be more beneficial in alleviating disease. Again, however, this requires more research but has been added into this review.
“The studies discussed above confirm the presence of aSyn and Lewy body-like deposits in individual cases of ALS and suggest a pathological role for aSyn expression in motor neuron degeneration. However, whether aSyn contributes to ALS pathology by exacerbating disease mechanisms, including mitochondrial dysfunction and oxidative stress, or whether aSyn and Lewy body pathology is a result of cellular processes associated with ALS disease is yet to be determined”
“Though further research is required to determine the mechanistic role of aSyn in ALS pathology, perhaps an aSyn-AO in combination therapy with an AO designed to target a known ALS hallmark will be beneficial in alleviating disease. Given the presence of SOD1 and TDP-43 pathology in PD and of aSyn and Tau pathology in ALS, therapies targeting each of these proteins may be utilised across the broader range of neurodegenerative disorders.”
- “ ~30% of ALS patients display Parkinsonian-like traits similar to those seen in PD and other synucleinopathies”. It indicated that AOs therapies targeting the alpha-synuclein may not be applied to all ALS treatments. So this review should clarify the scope of AOs therapies targeting the alpha-synuclein for ALS treatment.
Thank you to the reviewer for this suggestion. This idea in its entirety is novel and the literature isn’t there yet to define the scope of aSyn in ALS, however, we have further alluded to the research required in future studies to help define the scope and narrow down those patients amenable to an aSyn-targeted therapy at the end of the therapeutic section (section 5).
“The above strategies to reduce overall aSyn protein production have the potential to alleviate synucleinopathy disease pathology and resultant symptoms. Furthermore, given the overlap in the causative mechanisms of ALS, PD, and other synucleinopathies, these strategies may also be implemented into therapeutic practice to alleviate the disease progression of ALS. The current literature presented in this review demonstrates that aSyn has the potential to aggregate in ALS motor neurons, however, how broadly this is observed and the overall role of aSyn in ALS pathology is yet to be determined. Further studies of ALS patient cohorts investigating aSyn expression and Lewy body pathology will help to identify the targetable population and scope of patients amenable to an aSyn knockdown therapy.”
Reviewer 2 Report
In the review by Roberts et al., the authors explore the aetiology of ALS, how it might be a synucleinopathy and how RNA-based therapies may be potentially used as a treatment option. Overall, I found this review to be well written and topical, but it felt at times that synuclein and ALS were little disconnected in how the sections were written. In addition, some issues for me.
1. Often the authors refer to ALS as a synucleinopathy but were there ever any documented ALS cases associated with aggregated synuclein. Much of the evidence presented showed aggregates could form, but this might simply be incidental and a cause of dysfunction within other processes associated to ALS.
2. Also, beyond synuclein, what about Tau? Was this ever observed in ALS in its aggregated forms. FTD has the association with TDP43 and Tau, but could it also link ALS and PD.
3. While they do outline potential therapies for ALS and PD, is there a potential trial cohort in which this idea of ALS could be tested in which synuclein is targeted or is its aggregation simply a direct cause of the disease and mechanism that are driving its progression.
4. What about the converse, in PD are there hallmarks of misfolded TDP43 and SOD1 like in ALS, and could we apply a treatment for ALS in treating PD?
Author Response
Reviewer Two
Comments and Suggestions for Authors
In the review by Roberts et al., the authors explore the aetiology of ALS, how it might be a synucleinopathy and how RNA-based therapies may be potentially used as a treatment option. Overall, I found this review to be well written and topical, but it felt at times that synuclein and ALS were little disconnected in how the sections were written. In addition, some issues for me.
- Often the authors refer to ALS as a synucleinopathy but were there ever any documented ALS cases associated with aggregated synuclein. Much of the evidence presented showed aggregates could form, but this might simply be incidental and a cause of dysfunction within other processes associated to ALS.
To date, much of the evidence of alpha-synuclein involvement in ALS has looked at the coaggregation of alpha-synuclein with known ALS pathological proteins and not at the immediate effects of aSyn on ALS pathology. One study and its findings have however been added to this review that highlights the direct effects of aSyn aggregation on misfolding SOD1.
“Interestingly, one further study noted a significant increase in SOD1 oligomerisation following the introduction of misfolded aSyn preformed fibrils both in vitro and in vivo in hSOD1G93A mice. This study provides evidence that aSyn can induce or exacerbate aggregation of known ALS proteins, leading these authors to suggest that suppressing aSyn-mediated protein interaction may have a therapeutic potential for ALS.”
- Also, beyond synuclein, what about Tau? Was this ever observed in ALS in its aggregated forms. FTD has the association with TDP43 and Tau, but could it also link ALS and PD.
We thank the reviewer for this comment and agree that, yes whilst there is literature affirming the association of Tau and other proteins with known ALS proteins, this review focuses on the involvement of alpha-synuclein alone. We have now cleared this up in the review.
“Aggregations of further proteins linked to other neurodegenerative disease, such as the PD-linked alpha-synuclein (aSyn) and the AD-linked Tau, have been identified in ALS patient motor neurons, suggesting a role for these proteins in ALS pathology. This review, however, focuses on aSyn and the current literature surrounding its involvement in ALS pathogenesis.”
- While they do outline potential therapies for ALS and PD, is there a potential trial cohort in which this idea of ALS could be tested in which synuclein is targeted or is its aggregation simply a direct cause of the disease and mechanism that are driving its progression.
Though the approach to looking at alpha-synuclein as a treatment avenue in ALS is novel, we have now briefly addressed this issue in the concluding remarks of the review. Further comments added to this review address the need for future research to pinpoint the direct role of aSyn in ALS, however, the literature provided in this review clearly demonstrates that aSyn has the potential to aggregate in ALS moto neurons.
“Screening tools for genetic markers linked to disease, such as those structural variants associated with ALS, PD, and synucleinopathy described earlier, will be essential for identifying future clinical trial cohorts of patients likely to benefit from aSyn-targeting therapies.”
- What about the converse, in PD are there hallmarks of misfolded TDP43 and SOD1 like in ALS, and could we apply a treatment for ALS in treating PD?
We agree with the reviewer, that the potential for overlapping proteins and treatment options across neurodegenerative diseases is extremely interesting. This crossover between neurological disorders is exactly the avenue we are hoping to illuminate through this review. Presence of SOD1 and TDP-43 in PD is briefly addressed at the ends of sections 3.1 and in section 4 respectively and we have now tied this together at the end of the therapeutic section (section 5). However, extensive review of the literature regarding ALS hallmarks in PD is outside of the scope of this review.
“Though further research is required to determine the mechanistic role of aSyn in ALS pathology, perhaps an aSyn-AO in combination therapy with an AO designed to target a known ALS hallmark will be beneficial in alleviating disease. Given the presence of SOD1 and TDP-43 pathology in PD and of aSyn and Tau pathology in ALS, therapies targeting each of these proteins may be utilised across the broader range of neurodegenerative disorders.”